# ON THE ESTIMATION BIAS IN DOUBLE Q-LEARNING

## ABSTRACT

Double Q-learning is a classical method for reducing overestimation bias, which is caused by taking maximum estimated values in the Bellman operator. Its variants in the deep Q-learning paradigm have shown great promise in producing reliable value prediction and improving learning performance. However, as shown by prior work, double Q-learning is not fully unbiased and still suffers from underestimation bias. In this paper, we show that such underestimation bias may lead to multiple non-optimal fixed points under an approximated Bellman operation. To address the concerns of converging to non-optimal stationary solutions, we propose a simple and effective approach as a partial fix for underestimation bias in double Q-learning. This approach leverages real returns to bound the target value. We extensively evaluate the proposed method in the Atari benchmark tasks and demonstrate its significant improvement over baseline algorithms.

## 1 INTRODUCTION

Value-based reinforcement learning with neural networks as function approximators has become a widely-used paradigm and shown great promise in solving complicated decision-making problems in various real-world applications, including robotics control (Lillicrap et al., 2016), molecular structure design (Zhou et al., 2019), and recommendation systems (Chen et al., 2018). Towards understanding the foundation of these successes, investigating algorithmic properties of deep-learning-based value function approximation has been seen a growth of attention in recent years (Van Hasselt et al., 2018; Fu et al., 2019; Achiam et al., 2019; Dong et al., 2020). One of the phenomena of interest is that Q-learning (Watkins, 1989) is known to suffer from overestimation issues, since it takes a maximum operator over estimated action-values. Comparing with underestimated values, overestimation errors are more likely to be propagated through greedy action selections, which leads to an overestimation bias in value prediction (Thrun & Schwartz, 1993). This overoptimistic behavior of decision making has also been investigated in the literature of management science (Smith & Winkler, 2006) and economics (Thaler, 1988).

From a statistical perspective, the value estimation error may come from many sources, such as the stochasticity of the environment and the imperfection of the function expressivity. However, for deep Q-learning algorithms, even if most benchmark environments are nearly deterministic (Brockman et al., 2016) and millions of samples are collected, the overestimation phenomenon is still dramatic (Hasselt et al., 2016). One cause of this problematic issue is the difficulty of optimization. Although a deep neural network may have a sufficient expressiveness power to represent an accurate value function, the back-end optimization is hard to solve. As a result of computational considerations, stochastic gradient descent is almost the default choice for deep reinforcement learning algorithms. The high variance of such stochastic methods in gradient estimation would lead to an unavoidable approximation error in value prediction. This kind of approximation error cannot be addressed by simply increasing sample size and network capacity, which is a major source of overestimation bias.

Double Q-learning is a classical method to reduce the risk of overestimation, which is a specific variant of the double estimator (Stone, 1974) in the Q-learning paradigm. It uses a second value function to construct an independent action-value evaluation as cross validation. With proper assumptions, double Q-learning was proved to underestimate rather than overestimate the maximum expected values (Van Hasselt, 2010). However, in practice, obtaining two independent value estimators is usually intractable in large-scale tasks, which makes double Q-learning still suffer from overestimation in some situations. To address these empirical concerns, Fujimoto et al. (2018) proposed a variant named clipped double Q-learning, which takes the minimum over two value estimations.

This approach implicitly penalizes regions with high uncertainty (Fujimoto et al., 2019) and thus significantly repress the incentive of overestimation.

In this paper, we first review an analytical model adopted by prior work (Thrun & Schwartz, 1993; Lan et al., 2020) and reveal a fact that, due to the existence of underestimation biases, both double Q-learning and clipped double Q-learning have multiple approximated fixed points in this model. This result raises a concern that double Q-learning may easily get stuck in some local stationary regions and become inefficient in searching for the optimal policy. To bootstrap the ability of double Q-learning, we propose a simple heuristic that utilizes real return signals as a lower bound estimation to rule out the potential non-optimal fixed points. Benefiting from its simplicity, this method is easy to be combined with other existing techniques such as clipped double Q-learning. In the experiments on Atari benchmark tasks, we demonstrate that this simple approach is effective both in improving sample efficiency and convergence performance.

## 2 BACKGROUND

Markov Decision Process (MDP; Bellman, 1957) is a classical framework to formalize an agent-environment interaction system which can be defined as a tuple $\mathcal{M} = \langle \mathcal{S}, \mathcal{A}, P, R, \gamma \rangle$. We use $\mathcal{S}$ and $\mathcal{A}$ to denote the state and action space, respectively. $P(s'|s, a)$ and $R(s, a)$ denote the transition and reward functions, which are initially unknown to the agent. $\gamma$ is the discount factor. The goal of reinforcement learning is to construct a policy $\pi : \mathcal{S} \to \mathcal{A}$ maximizing discounted cumulative rewards,

$$V^\pi(s) = \mathbb{E}\left[\sum_{t=0}^\infty \gamma^t R(s_t, \pi(s_t)) \middle| s_0 = s, s_{t+1} \sim P(\cdot|s_t, \pi(s_t))\right].$$

Another quantity of interest in policy learning can be defined through the Bellman equation $Q^\pi(s, a) = R(s, a) + \gamma \mathbb{E}_{s' \sim P(\cdot|s,a)}[V^\pi(s')]$. The optimal value function $Q^*$ corresponds to the unique solution of the Bellman optimality equation,

$$\forall (s, a) \in \mathcal{S} \times \mathcal{A}, \ Q^*(s, a) = R(s, a) + \gamma \mathop{\mathbb{E}}_{s' \sim P(\cdot|s,a)}\left[\max_{a' \in \mathcal{A}} Q^*(s', a')\right].$$

Q-learning algorithms are based on the Bellman operator $\mathcal{T}$ stated as follows:

$$(\mathcal{T}Q)(s, a) = R(s, a) + \gamma \mathop{\mathbb{E}}_{s' \sim P(\cdot|s,a)}\left[\max_{a' \in \mathcal{A}} Q(s', a')\right]. \tag{1}$$

By iterating this operator, value iteration is proved to converge to the optimal value function $Q^*$. To extend Q-learning methods to real-world applications, function approximation is indispensable to deal with a high-dimensional state space. Deep Q-learning (Mnih et al., 2015) considers a sample-based objective function and constructs an iterative optimization framework:

$$\theta_{t+1} \leftarrow \arg\min_{\theta \in \Theta} \mathop{\mathbb{E}}_{(s,a,r,s') \sim \mathcal{D}}\left[\left(r + \gamma \max_{a' \in \mathcal{A}} Q_{\theta_t}(s', a') - Q_\theta(s, a)\right)^2\right], \tag{2}$$

in which $\Theta$ denotes the parameter space of the value network, and $\theta_0 \in \Theta$ is initialized by some predetermined method. $\mathcal{D}$ is the data distribution which is changing during exploration. With infinite samples and a sufficiently rich function class, the update rule stated in Eq. (2) is asymptotically equivalent to applying the Bellman operator $\mathcal{T}$, but the underlying optimization is usually inefficient in practice. In deep Q-learning, Eq. (2) is optimized by mini-batch gradient descent and thus its value estimation suffers from unavoidable approximation errors.

## 3 EFFECTS OF UNDERESTIMATION BIAS IN DOUBLE Q-LEARNING

In this section, we will first review a common analytical model used by previous work studying estimation bias (Thrun & Schwartz, 1993; Lan et al., 2020), in which double Q-learning is known to have underestimation bias. Based on this analytical model, we show that its underestimation bias could make double Q-learning have multiple fixed point solutions under an approximated Bellman operation. This result suggests that double Q-learning may have extra non-optimal stationary solutions under the effects of the approximation error.

## 3.1 MODELING APPROXIMATION ERROR IN Q-LEARNING

Following Thrun & Schwartz (1993) and Lan et al. (2020), we formalize the underlying approximation error $e^{(t)}(s, a)$ of target value regression as a set of random noises impacting on the Bellman operation,

$$Q^{(t+1)}(s, a) = (\widetilde{\mathcal{T}} Q^{(t)})(s, a) = (\mathcal{T} Q^{(t)})(s, a) + e^{(t)}(s, a), \tag{3}$$

where $\mathcal{T}$ denotes the ground truth Bellman operator using full information of the MDP (see Eq. (1)), and $\widetilde{\mathcal{T}}$ denotes a stochastic operator with noisy outputs. The main purpose of introducing the explicit noise term $e^{(t)}(s, a)$ is to emphasize that the approximation error discussed here is different from the sampling error. In an information-theoretic perspective, the sampling error can be reduced asymptotically as the sample size increases. However, there is a barrier of optimization difficulty to establish a precise estimation in practice, which leads to an unavoidable approximation error in optimization. By integrating the noise term $e^{(t)}(s, a)$ into the Bellman operation, we set up an analytical model to investigate how Q-learning algorithms interact with the inherent approximation error.

In this model, double Q-learning (Van Hasselt, 2010) can be modeled by two estimator instances $\{Q_i^{(t)}\}_{i \in \{1,2\}}$ with separated noise variables $\{e_i^{(t)}\}_{i \in \{1,2\}}$. For simplification, we introduce a policy function $\pi^{(t)}(s) = \arg\max_a Q_1^{(t)}(s, a)$ to override the state value function as follows:

$$\forall i \in \{1, 2\}, \quad Q_i^{(t+1)}(s, a) = R(s, a) + \gamma \mathop{\mathbb{E}}_{s' \sim P(\cdot|s,a)} \left[ V^{(t)}(s') \right] + e_i^{(t)}(s, a),$$

$$V^{(t)}(s) = Q_2^{(t)}(s, \pi(s)) = Q_2^{(t)} \left( s, \ \arg\max_{a \in \mathcal{A}} Q_1^{(t)}(s, a) \right). \tag{4}$$

The difference of Eq. (4) from the definition of double Q-learning given by Van Hasselt (2010) is using a unified target value $V^{(t)}(s')$ for both two estimators. This simplification does not affect the derived implications of double Q-learning, and is also implemented by advanced variants of double Q-learning (Fujimoto et al., 2018; Lan et al., 2020).

Note that the target value can be constructed only using the state-value function. Based on this observation, we can define the fixed point of a stochastic operators on the target value.

**Definition 1** (Approximated Fixed Points). *Let $\widetilde{\mathcal{T}}$ denote a stochastic Bellman operator, such as what are stated in Eq. (3) and Eq. (4). A state-value function $V$ is regarded as an approximated fixed point under a stochastic Bellman operator $\widetilde{\mathcal{T}}$ if it satisfies $\mathbb{E}[\widetilde{\mathcal{T}} V] = V$, where $\widetilde{\mathcal{T}} V$ denotes the output state-value function while applying the Bellman operator $\widetilde{\mathcal{T}}$ on $V$.*

The fixed point defined above is named *approximated* since they are not truly static, but is invariant under the stochastic Bellman operation in expectation. In Appendix A.2, we will prove the existence of such fixed points as the following statement.

**Proposition 1.** *Assume the probability density functions of the noise terms $\{e(s, a)\}$ are continuous. The stochastic Bellman operators defined by Eq. (3) and Eq. (4) have approximated fixed points defined as Definition 1.*

## 3.2 EXISTENCE OF MULTIPLE APPROXIMATED FIXED POINTS IN DOUBLE Q-LEARNING

Given the definition of the approximated fixed point, a natural question is whether such kind of fixed points are unique or not. Recall that the optimal value function $Q^*$ is the unique solution of the Bellman optimality equation, which is the foundation of Q-learning algorithms. However, in this section, we will show that, under the effects of the approximation error, the approximated fixed points of double Q-learning may not be unique.

Figure 1a illustrates a simple MDP in which double Q-learning stated as Eq. (4) has multiple approximated fixed points. This MDP is fully deterministic and contains only two states $s_0$ and $s_1$. All actions in state $s_1$ lead to a self-loop and produce a unit reward signal. On state $s_0$, the result of executing action $a_0$ is a self-loop with a slightly larger reward signal than choosing action $a_1$ which leads to state $s_1$. The only challenge for decision making in this MDP is to distinguish the outcomes of executing action $a_0$ and $a_1$ on state $s_0$. To make the example more accessible, we assume the approximation errors $\{e^{(t)}(s, a)\}_{t,s,a}$ are a set of independent random variables following a uniform

$R_{0,0}=1.1 \quad R_{1,0}=R_{1,1}=1$

$s_0 \xrightarrow{R_{0,1}=1} s_1$

$\epsilon = 1.0 \quad \gamma = 0.99$

(a) A simple construction

| $V(s_0)$ | $V(s_1)$ | $\tilde{\pi}(a_0|s_0)$ |
|---|---|---|
| 100.162 | 100.0 | 62.2% |
| 101.159 | 100.0 | 92.9% |
| 110.0 | 100.0 | 100.0% |

(b) Numerical solutions of fixed points

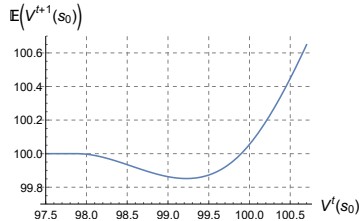

(c) Visualizing non-monotonicity

Figure 1: (a) A simple infinite-horizon MDP where double Q-learning stated as Eq. (4) has multiple approximated fixed points. $R_{i,j}$ is a shorthand of $R(s_i, a_j)$. (b) The numerical solutions of the fixed points produced by double Q-learning in the MDP presented above. $\tilde{\pi}$ refers to Definition 3. (c) The relation between the input state-value $V^{(t)}(s_0)$ and the expected output state-value $\mathbb{E}[V^{(t+1)}(s_0)]$ generated by double Q-learning in the constructed MDP, in which we assume $V^{(t)}(s_1) = 100$.

distribution $Uniform(-\epsilon, \epsilon)$. This simplification is also adopted by Thrun & Schwartz (1993) and Lan et al. (2020) in case studies. Here, we select the magnitude of noise as $\epsilon = 1.0$ and the discount factor as $\gamma = 0.99$ to balance the scale of involved amounts.

Considering to solve the equation $\mathbb{E}[\widetilde{\mathcal{T}}V] = V$ according to the definition of the approximated fixed point (see Definition 1), the numerical solutions of such fixed points are presented in Table 1b. There are three different fixed point solutions. The first thing to notice is that the optimal fixed point $V^*$ is retained in this MDP (see the last row of Table 1b), since the noise magnitude $\epsilon = 1.0$ is much smaller than the optimality gap $Q^*(s_0, a_0) - Q^*(s_0, a_1) = 10$. The other two fixed points are non-optimal and very close to $Q(s_0, a_0) \approx Q(s_0, a_1) = 100$, in which the agent cannot fully distinguish the difference between choosing $a_0$ and $a_1$. To illustrate this example, we first present a sufficient condition for a stochastic Bellman operator to have multiple fixed points, and then we give an intuitive explanation to connect this condition with the constructed MDP.

**Mathematical Condition.** Note that the definition of the stochastic Bellman operator is a model of an imprecise target value regression. From this perspective, the input of a stochastic Bellman operator can be defined as a set of ground truth target values $\{(\mathcal{T}Q^{(t)})(s,a)\}_{s,a}$. Based on this notation, a sufficient condition for the existence of multiple fixed points is stated as follows.

**Proposition 2.** *Let $f_s\left(\{(\mathcal{T}Q)(s,a)\}_{a\in\mathcal{A}}\right) = \mathbb{E}[(\widetilde{\mathcal{T}}V)(s)]$ denote the expected output value of the stochastic Bellman operator $\widetilde{\mathcal{T}}$ on state $s$, and assume $f_s(\cdot)$ is differentiable. If a stochastic Bellman operator $\widetilde{\mathcal{T}}$ satisfies Eq. (5), there exists an MDP such that $\widetilde{\mathcal{T}}$ has multiple fixed points.*

$$\exists i, \exists X \in \mathbb{R}^{|\mathcal{A}|}, \quad \frac{\partial}{\partial x_i} f_s(X) > 1, \tag{5}$$

*where $X = \{x_i\}_{i=1}^{|\mathcal{A}|}$ denotes the input of the function $f_s$.*

The proof of Proposition 2 is deferred to Appendix A.4. This proposition suggests that, in order to determine whether a given stochastic Bellman operator $\widetilde{\mathcal{T}}$ may have multiple fixed points, we need to check whether its expected output values could change dramatically with a slight alter of the input values.

**Intuitive Explanation.** To provide the intuition for understanding the constructed MDP in Figure 1, we introduce a simple property of a Bellman operator, named monotonicity, which helps to illustrate how the sufficient condition in Eq. (5) are met in double Q-learning.

**Definition 2** (Monotonicity). *A stochastic Bellman operator $\widetilde{\mathcal{T}}$ is called approximately monotonic if it satisfies the following property:*

$$\forall \mathcal{M}, \forall V_1, V_2 \in \mathbb{R}^{\mathcal{S}}, \quad (\forall s \in \mathcal{S}, V_1(s) \geq V_2(s)) \Rightarrow \left(\forall s \in \mathcal{S}, \mathbb{E}[(\widetilde{\mathcal{T}}V_1)(s)] \geq \mathbb{E}[(\widetilde{\mathcal{T}}V_2)(s)]\right).$$

Although this monotonicity property holds for vanilla Q-learning (Watkins, 1989) and a recent variant called maxmin Q-learning (Lan et al., 2020), it absents from double Q-learning (Van Hasselt, 2010) and a famous variant called clipped double Q-learning (Fujimoto et al., 2018). The monotonicity property is not a sufficient condition for Eq. (5) but it is quite important for understanding the constructed example in Figure 1.

Considering the constructed MDP as an example, Figure 1c visualizes the relation between the input state-value $V^{(t)}(s_0)$ and the expected output state-value $\mathbb{E}[V^{(t+1)}(s_0)]$ while assuming $V^{(t)}(s_1) = 100$ has converged to its stationary point. The minima point of the output value is located at the situation where $V^{(t)}(s_0)$ is slightly smaller than $V^{(t)}(s_1)$, since the expected policy derived by $\widetilde{\mathcal{T}}V^{(t)}$ will have a remarkable probability to choose sub-optimal actions. This local minima suffers from the most dramatic underestimation among the whole curve, and the underestimation will eventually vanish as the value of $V^{(t)}(s_0)$ increases. During this process, a large magnitude of the first-order derivative could be found to meet the condition stated in Eq. (5).

A heuristics to verify the condition in Eq. (5) is to check the nearby regions of the local minima and maxima in a non-monotonic function $\mathbb{E}[\widetilde{\mathcal{T}}V]$. In Appendix A.5, we also show that clipped double Q-learning, another method with non-monotonicity, has multiple fixed points in an MDP slightly modified from Figure 1a.

**Implications**  To facilitate discussions, we introduce a concept, named induced policy, to characterize how the agent behaves and evolves around these approximated fixed points.

**Definition 3** (Induced Policy). *Given a target state-value function $V$, its induced policy $\tilde{\pi}$ is defined as a stochastic action selection according to the value estimation produced by a stochastic Bellman operation,*

$$\tilde{\pi}(a|s) = \mathbb{P}\left[ a = \arg\max_{a' \in \mathcal{A}} \left( \underbrace{R(s,a') + \gamma \mathop{\mathbb{E}}_{s' \sim P(\cdot|s,a')} [V(s')]}_{(\mathcal{T}Q)(s,a')} + e_1(s,a') \right) \right],$$

*where $\{e_1(s,a)\}_{s,a}$ are drawing from the same noise distribution as what is used by double Q-learning stated in Eq. (4).*

**Proposition 3.** *Assume the noise terms $e_1$ and $e_2$ are independently generated in the double estimator stated in Eq. (4). Every approximated fixed point $V$ is equal to the ground truth value function $V^{\tilde{\pi}}$ with respect to an induced policy $\tilde{\pi}$.*

The proof of Proposition 3 is deferred to Appendix A.3. As shown by this proposition, the estimated value of a non-optimal fixed point is corresponding to the value of a stochastic policy, which revisits the incentive of double Q-learning to underestimate true maximum values. Based on Proposition 3, the concept of induced policy can provide a snapshot to infer the algorithmic properties of approximated fixed points. Taking the third column of Table 1b as an example, due to the existence of the approximation error, the induced policy $\tilde{\pi}$ suffers from a remarkable uncertainty in determining the best action on state $s_0$. This phenomenon suggests that the approximation error would make the greedy action selection become competitive among actions with near-maximum values. Once the approximation error leads the policy to select a non-optimal action, the learning of the value function would get stuck in non-optimal fixed points. Note that, according to the characterization given by Definition 3, the stationary property of such kind of fixed points might be similar to that of saddle points in the literature of optimization. Whether the agent gets stuck in non-optimal solutions depends on the actual behavior of approximation error. In section 4, we will introduce a method to provide incentives for escaping non-optimal fixed points by reducing underestimation bias.

### 3.3 APPROXIMATION ERROR IN DEEP Q-LEARNING

In the MDP example presented in the previous section, the noise terms $\{e^{(t)}(s,a)\}_{t,s,a}$ are modeled by a set of uniform random variables $Uniform(-\epsilon, \epsilon)$. The noise magnitude $\epsilon = 1.0$ seems to a bit large when it compares with the scale of one-step reward signal, but it is actually quite small in terms of the scale of the value function. Considering an MDP only containing positive rewards, the scale of value functions would be in $[0, R_{\max}/(1-\gamma)]$ where $R_{\max}$ denotes the upper bound of reward signals. In this setting, if the relative error of solving the target value regression achieves the order of $1/(1-\gamma)$, the induced approximation error will reach the same scale of reward signals. Revisiting the MDP introduced in the previous section with the

Table 1: Evaluating the mean absolute TD-error of DDQN in WizardOfWor.

| samples | $|Q|$ | TD-error |
|---------|-------|----------|
| 3M      | 0.310 | 0.014    |
| 5M      | 0.303 | 0.014    |
| 10M     | 0.471 | 0.015    |

discount factor $\gamma = 0.99$, the noise magnitude $\epsilon = 1.0$ is only an $1\%$ relative error comparing to the scale of the value function, which finally leads to a $10\%$ relative error in value estimation. It is worth noticing that such a small approximation error can cause significantly sub-optimal fixed points. Table 1 presents the scale of the estimation error of DDQN in a deterministic version of one Atari benchmark task. We evaluate the average absolute TD-error $\mathbb{E}[|Q - \mathcal{T}\hat{Q}|]$ using a batch of transitions sampling from the replay buffer. The evaluation is processed before the target value switches. As shown in Table 1, the value estimation of DDQN suffers from a constant approximation error, which may cause the potential risk of multiple non-optimal fixed points according to our analysis.

## 4    LOWER BOUNDED DOUBLE Q-LEARNING

As discussed in the last section, the underestimation bias of double Q-learning may lead to multiple non-optimal fixed points in the analytical model. A major source of such underestimation is the inherent approximation error caused by the difficulty of optimization. In this section, we will introduce a simple heuristic, named lower bounded double Q-learning, which helps to reduce the negative effects of underestimation.

The main idea is motivated by a fact that the value prediction produced by double Q-learning does not tend to overestimate the true maximum value. It is a major difference from the overestimation bias of vanilla Q-learning. When the environment is deterministic, the return values of collected trajectories can naturally serve as a lower bound for value estimation. To utilize this lower bound, we modify the objective function of deep Q-learning as follows,

$$L(\theta) = \mathop{\mathbb{E}}_{(s_t,a_t,r_t,s_{t+1}) \sim \mathcal{D}} \left[ \left( r_t + \gamma \max \left( V_{\hat{\theta}}(s_{t+1}), V_\tau(s_{t+1}) \right) - Q_\theta(s_t, a_t) \right)^2 \right],$$

where the target value is computed by taking the maximum over two sources of estimation. The first term $V_{\hat{\theta}}(s_{t+1})$ is the target state-value computed by the frozen parameter $\hat{\theta}$. The second term $V_\tau(s_{t+1})$ is the discounted return value of the corresponding trajectory $\tau$ in the replay buffer. Formally, $V_\tau(s_{t+1}) = \sum_{k=0}^{H-t} \gamma^k r_{t+k+1}$ where $H$ denotes the length of the trajectory.

This lower bounded objective has three potential impacts on the value estimation:

- Considering the non-optimal fixed points of double Q-learning discussed in the previous section, the values of these fixed points are corresponding to some sub-optimal polices which underestimate the optimal action-values (see Proposition 3). Under the effects of exploration, some of the collected trajectories may have higher returns than the current policy, which would provide an incentive to get rid of the non-optimal fixed points through the lower bounded objective.

- When the return signal $V_\tau$ has approached the ground truth maximum value $V^*$, the lower bounded objective will lead to an overestimation bias, since only underestimated values are cut off by the maximum operator. In practice, the effects of such overestimation bias are manageable, because all trajectories are collected by an exploratory policy which can hardly produce the ground truth maximum values.

- In addition, when the environment carries some extent of stochasticity, the trajectories with high return values would cause additional risks of overestimation through the proposed lower bounded objective. In the other hand, our method excels at addressing the issue of non-optimal fixed points, which is exacerbated by the environment stochasticity. This is because the environment stochasticity increases the difficulty of optimization in the target regression and leads to larger approximation error that generates non-optimal fixed points. From this perspective, our method is a trade-off between the underestimation bias of double Q-learning and the overestimation of the trajectory-based lower-bound estimation. The effectiveness of such trade-off will investigated empirically in section 5.

In this paper, we investigate two methods to compute the estimated target value $V_{\hat{\theta}}(s_{t+1})$. The first choice $V_{\hat{\theta}}^{\text{DDQN}}(s')$ corresponds to the implementation proposed by double deep Q-network (DDQN;

Hasselt et al., 2016),

$$V_{\hat{\theta}}^{\text{DDQN}}(s') = Q_{\hat{\theta}}\left(s', \arg\max_{a'\in\mathcal{A}} Q_\theta(s', a')\right).$$

It is an adaption of double Q-learning to the framework of deep Q-network (DQN; Mnih et al., 2015), in which the action selection of target values are produced by the online parameter $\theta$.

An imperfection of the combination with DDQN is that, DDQN is known to suffer from overestimation sometimes since value predictions produced by $\theta$ and $\hat{\theta}$ are not fully independent. When the overestimation happens, the real return signal $V_\tau(s_{t+1})$ cannot provide any benefits to the value prediction. To address these concerns, we also adopt another advanced technique called clipped double Q-learning (Fujimoto et al., 2018):

$$V_{\hat{\theta}}^{\text{CDDQN}}(s') = \min_{i\in\{1,2\}} Q_{\hat{\theta}_i}\left(s', \arg\max_{a'\in\mathcal{A}} Q_{\hat{\theta}_1}(s', a')\right).$$

This formulation of computing target values is adapted from its actor-critic version, and we name it clipped double deep Q-network (Clipped DDQN). This method uses a shared objective function to optimize two copies of value networks $Q_{\theta_1}$ and $Q_{\theta_2}$, in which the target values are computed by taking the minimum over two separated estimations $Q_{\hat{\theta}_1}$ and $Q_{\hat{\theta}_2}$. This minimum operator will introduces an underestimation bias (Ciosek et al., 2019) which significantly repress the incentive of overestimation in all sources. This underestimated estimator is appropriate to combine with the lower bounded objective since the lower bound provide by trajectory returns can help to prevent the underestimation error from propagation.

## 5 EXPERIMENTS

Our experiment environments are based on the Atari benchmark tasks in OpenAI Gym (Brockman et al., 2016). All baselines and our approaches are implemented using the same set of hyperparameters. A detailed description of experiment settings is deferred to Appendix B.

**Overall performance comparison.** We investigated six variants of deep Q-learning algorithms, including DQN (Mnih et al., 2015), double DQN (DDQN; Hasselt et al., 2016), dueling DDQN (Wang et al., 2016), averaged DQN (Anschel et al., 2017), maxmin DQN (Lan et al., 2020), and clipped double DQN adapted from Fujimoto et al. (2018). Our proposed lower bounded objective is built upon two variants of double Q-learning which have clear incentive of underestimation. As shown in Figure 2, the proposed lower bounded objective has great promise in bootstrapping the performance of double Q-learning algorithms. The improvement can be observed both in terms of sample efficiency and final performance. This advance may credit to the removal of potential non-optimal fixed points. Another notable observation is that, the effects of adopting clipped double Q-learning are environment-dependent, since it has an underestimation bias which cannot be fully eliminated by the lower bounded objective. In Appendix C, we present more experimental results to reveal the superiority of the proposed method.

**Comparison with $n$-step bootstrapping.** In addition to our proposed lower bounded objective, $n$-step bootstrapping (Sutton, 1988) also has potential to attenuate the issue of non-optimal fixed points by utilizing real returns. An advantage of our method is that the performance of $n$-step bootstrapping is sensitive to the choice of the hyper-parameter $n$. In Figure 3, we evaluate the performance of DDQN with different values of $n$ chosen by Hessel et al. (2018). The results indicate that our proposed lower bounded objective is compatible with $n$-step bootstrapping. The performance can be

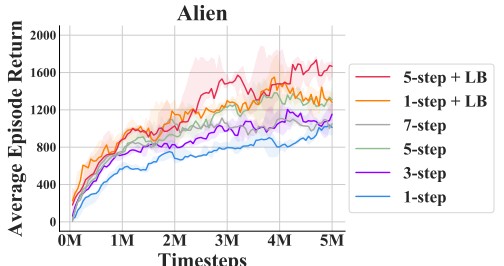

Figure 3: Learning curves of DDQN with $n$-step bootstrapping.

further improved by combing two approaches. More experimental results on the comparison with $n$-step bootstrapping are provided in Appendix D.

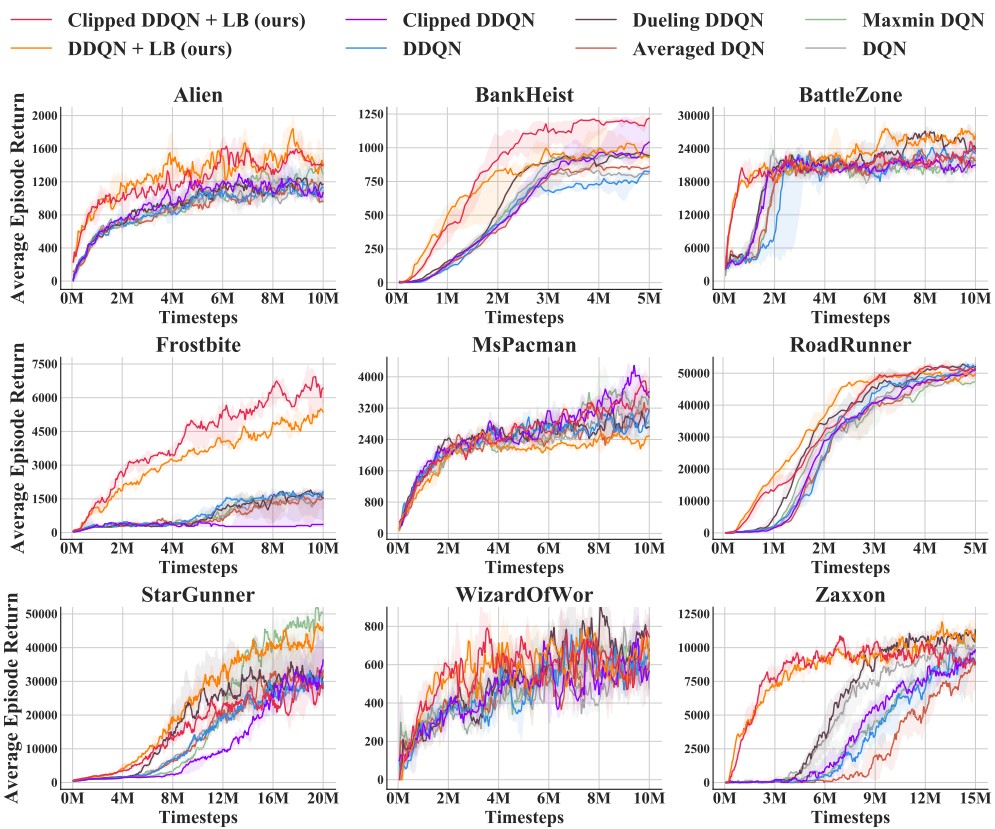

Figure 2: Learning curves on a suite of Atari benchmark tasks, in which the "+LB" version corresponds to the combination with the proposed lower bounded objective. All curves presented in this paper are plotted from the median performance over 5 runs with random initialization. To make the comparison more clear, the curves are smoothed by averaging 10 most recent checkpoints. More experiment results are included in Appendix C.

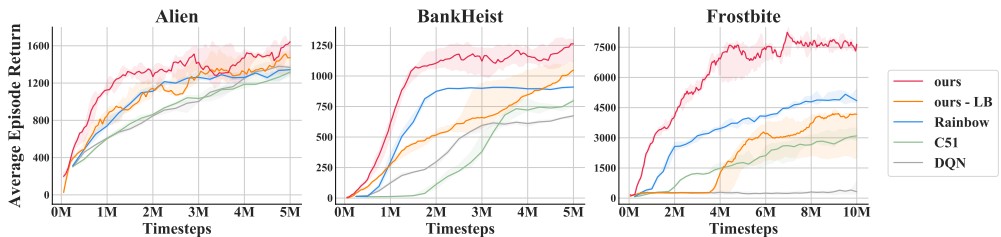

Figure 4: Learning curves on a suite of stochastic Atari benchmark tasks with sticky actions, in which "ours" version corresponds to the combination of the proposed lower bounded objective with clipped double Q-learning and multi-step bootstrapping. "ours - LB" corresponds to the ablation study without the lower bounded objective. The curves of Rainbow, C51, and DQN are released by Castro et al. (2018) using the same environment setting.

**Performance comparison in stochastic environments with sticky actions** The experiments presented in Figure 2 and Figure 3 are based on the standard environment setting used by Mnih et al. (2015), in which the environment stochasticity only comes from the partial observability of visual observations. When the environment is stochastic, the trajectories with high return values would cause additional risks of overestimation through the proposed lower bounded objective. To investigate the performance of our method in stochastic environments, we adopt a stochastic variant of Atari environment setting with sticky actions Machado et al. (2018), in which every environ-

ment step will execute the agent's previous action with probability 0.25. In this experiment, we investigate two state-of-the-art baselines, Rainbow (Hessel et al., 2018) and C51 (Bellemare et al., 2017), which use distributional representation to address the optimization difficulty of environment stochasticity. To fairly compare with these state-of-the-art baselines that integrates many standard techniques, we also combine our proposed lower bounded objective with clipped double Q-learning and multi-step bootstrapping. As shown in Figure 4, our proposed method still outperform baselines in stochastic environments. This result suggests that, comparing to the overestimation bias introduced by the lower bounded objective, stuck in non-optimal fixed points is a more critical issue for deep Q-learning algorithms. Another explanation is that the environment stochasticity may increase the difficulty of optimization and lead to larger approximation error for generating non-optimal fixed points. Our proposed method is an effective trade-off between the risks of overestimation and non-optimal fixed points.

## 6   RELATED WORK

Correcting the estimation bias in double Q-learning is an active topic which induces a series of approaches. Weighted double Q-learning (Zhang et al., 2017) considers an importance weight parameter to integrate the overestimated and underestimated estimators. Clipped double Q-learning (Fujimoto et al., 2018), which uses a minimum operator in target values, has become the default implementation of most advanced actor-critic algorithms (Haarnoja et al., 2018). Based on clipped double Q-learning, several methods have been investigated to reduce the its underestimation and achieve promising performance Ciosek et al. (2019); Li & Hou (2019).

Besides the variants of double Q-learning, there are lots of other techniques proposed regarding the bias-variance trade-off in Q-learning algorithms. Similar to our proposed approach, most of existing methods focus on the construction of target values. Averaged DQN (Anschel et al., 2017) uses multiple historical copies of target networks to reduce the variance of estimation. The target used by truncated quantile critics (TQC; Kuznetsov et al., 2020) is based on a distributional representation of value functions, which contributes to reduce the overestimation of actor-critic methods. Using the softmax operator in Bellman operation is also considered as an effective approach to reduce the effects of approximation error (Fox et al., 2016; Asadi & Littman, 2017; Song et al., 2019; Kim et al., 2019).

The characteristic of our approach is the usage of real return signals, which uses the environment prior to break statistical barriers. Our method is a variant of self-imitation (Oh et al., 2018) and is integrated with the Bellman operation. The effectiveness of such a ground truth lower bound is also observed by a recent work (Fujita et al., 2020). When the environment carries stochasticity, taking the average return among similar states is a plausible approach to reducing the bias of the trajectory-based estimation. From this perspective, the proposed method is also related to the memory-based approaches for nearly deterministic environments, such as episodic control (Blundell et al., 2016; Pritzel et al., 2017) and graph-based planning (Huang et al., 2019; Eysenbach et al., 2019).

## 7   CONCLUSION

In this paper, we reveal an interesting fact that, under the effects of approximation error, double Q-learning may have multiple non-optimal fixed points. The main cause of such non-optimal fixed points is the underestimation bias of double Q-learning. Regarding this issue, we provide some analysis to characterize what kind of Bellman operators may suffer from the same problem, and how the hard cases are constructed. To address the potential risk of converging to non-optimal solutions, we propose an alternative objective function to reduce the underestimation in double Q-learning. The main idea of this approach is to leverage real environment returns as a lower bound for value estimation. Our proposed method is simple and easy to be combined with other advanced techniques in deep Q-learning. The experiments show that the proposed method has shown great promise in improving both sample efficiency and convergence performance. It achieves a significant improvement over baselines algorithms on Atari benchmark environments.

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

## A OMITTED STATEMENTS AND PROOFS

### A.1 THE RELATION BETWEEN ESTIMATION BIAS AND APPROXIMATED FIXED POINTS

An intuitive characterization of such fixed point solutions is considering one-step estimation bias with respect to the maximum expected value, which is defined as

$$\mathcal{E}(\widetilde{\mathcal{T}}, V, s) = \mathbb{E}[(\widetilde{\mathcal{T}}V)(s)] - (\mathcal{T}V)(s), \tag{6}$$

where $(\mathcal{T}V)(s)$ corresponds to the precise state value after applying the ground truth Bellman operation. The amount of estimation bias $\mathcal{E}$ characterizes the deviation from the standard Bellman operator $\mathcal{T}$, which can be regarded as imaginary rewards in fixed point solutions.

Every approximated fixed point solution under a stochastic Bellman operator can be characterized as the optimal value function in a modified MDP where only the reward function is changed.

**Proposition 4.** *Let $\widetilde{V}$ denote an approximation fixed point under a stochastic Bellman operator $\widetilde{\mathcal{T}}$. Define a modified MDP $\widetilde{\mathcal{M}} = \langle \mathcal{S}, \mathcal{A}, P, R + \widetilde{R}, \gamma \rangle$ based on $\mathcal{M}$, where the extra reward term is defined as*

$$\widetilde{R}(s, a) = \mathcal{E}(\widetilde{\mathcal{T}}, \widetilde{V}, s) = \mathbb{E}[(\widetilde{\mathcal{T}}\widetilde{V})(s)] - (\mathcal{T}\widetilde{V})(s),$$

*where $\mathcal{E}$ is the one-step estimation bias defined in Eq. (6). Then $\widetilde{V}$ is the optimal state-value function of the modified MDP $\widetilde{\mathcal{M}}$.*

*Proof.* Define a value function $\widetilde{Q}$ based on $\widetilde{V}$, $\forall(s, a) \in \mathcal{S} \times \mathcal{A}$,

$$\widetilde{Q}(s, a) = R(s, a) + \widetilde{R}(s, a) + \mathop{\mathbb{E}}_{s' \sim P(\cdot|s,a)} [\widetilde{V}(s')].$$

We can verify $\widetilde{Q}$ is consistent with $\widetilde{V}$, $\forall s \in \mathcal{S}$,

$$\begin{aligned}
\widetilde{V}(s) &= \mathbb{E}[(\widetilde{\mathcal{T}}\widetilde{V})(s)] \\
&= \mathbb{E}[(\widetilde{\mathcal{T}}\widetilde{V})(s)] - (\mathcal{T}\widetilde{V})(s) + \max_{a \in \mathcal{A}} \left( R(s, a) + \gamma \mathop{\mathbb{E}}_{s' \sim P(\cdot|s,a)} [\widetilde{V}(s')] \right) \\
&= \max_{a \in \mathcal{A}} \left( R(s, a) + \widetilde{R}(s, a) + \gamma \mathop{\mathbb{E}}_{s' \sim P(\cdot|s,a)} [\widetilde{V}(s')] \right) \\
&= \max_{a \in \mathcal{A}} \widetilde{Q}(s, a).
\end{aligned}$$

Let $\mathcal{T}_{\widetilde{\mathcal{M}}}$ denote the Bellman operator of $\widetilde{\mathcal{M}}$. We can verify $\widetilde{Q}$ satisfies Bellman optimality equation to prove the given statement, $\forall(s, a) \in \mathcal{S} \times \mathcal{A}$,

$$\begin{aligned}
(\mathcal{T}_{\widetilde{\mathcal{M}}}\widetilde{Q})(s, a) &= R(s, a) + \widetilde{R}(s, a) + \gamma \mathop{\mathbb{E}}_{s' \sim P(\cdot|s,a)} \left[ \max_{a' \in \mathcal{A}} \widetilde{Q}(s', a') \right] \\
&= R(s, a) + \widetilde{R}(s, a) + \gamma \mathop{\mathbb{E}}_{s' \sim P(\cdot|s,a)} [\widetilde{V}(s')] \\
&= \widetilde{Q}(s, a).
\end{aligned}$$

Thus we can see $\widetilde{V}$ is the solution of Bellman optimality equation in $\widetilde{\mathcal{M}}$. □

### A.2 THE EXISTENCE OF APPROXIMATED FIXED POINTS

The key technique for proving the existence of approximated fixed points is Brouwer's fixed point theorem.

**Lemma 1.** *Let $B = [-L, -L]^d$ denote a $d$-dimensional bounding box. For any continuous function $f : B \to B$, there exists a fixed point $x$ such that $f(x) = x \in B$.*

*Proof.* It refers to a special case of Brouwer's fixed point theorem (Brouwer, 1911). $\quad\square$

**Lemma 2.** *Let $\widetilde{\mathcal{T}}$ denote the stochastic Bellman operator defined by Eq. (3). There exists a real range $L$, $\forall V \in [L, -L]^{|\mathcal{S}|}$, $\mathbb{E}[\widetilde{\mathcal{T}}V] \in [L, -L]^{|\mathcal{S}|}$.*

*Proof.* Let $R_{\max}$ denote the range of the reward function for MDP $\mathcal{M}$. Let $R_e$ denote the range of the noisy term. Formally,

$$R_{\max} = \max_{(s,a) \in \mathcal{S} \times \mathcal{A}} |R(s, a)|,$$

$$R_e = \max_{s \in \mathcal{S}} \mathbb{E}\left[\max_{a \in \mathcal{A}} |e(s, a)|\right].$$

Note that the $L_\infty$-norm of state value functions satisfies $\forall V \in \mathbb{R}^{|\mathcal{S}|}$,

$$\|\mathbb{E}[\widetilde{\mathcal{T}}V]\|_\infty \leq R_{\max} + R_e + \gamma \|V\|_\infty.$$

We can construct the range $L = (R_{\max} + R_e)/(1 - \gamma)$ to prove the given statement. $\quad\square$

**Lemma 3.** *Let $\widetilde{\mathcal{T}}$ denote the stochastic Bellman operator defined by Eq. (4). There exists a real range $L$, $\forall V \in [L, -L]^{|\mathcal{S}|}$, $\mathbb{E}[\widetilde{\mathcal{T}}V] \in [L, -L]^{|\mathcal{S}|}$.*

*Proof.* Let $R_{\max}$ denote the range of the reward function for MDP $\mathcal{M}$. Formally,

$$R_{\max} = \max_{(s,a) \in \mathcal{S} \times \mathcal{A}} |R(s, a)|.$$

Note that the $L_\infty$-norm of state value functions satisfies $\forall V \in \mathbb{R}^{|\mathcal{S}|}$,

$$\|\mathbb{E}[\widetilde{\mathcal{T}}V]\|_\infty \leq R_{\max} + \gamma \|V\|_\infty.$$

We can construct the range $L = R_{\max}/(1 - \gamma)$ to prove the given statement. $\quad\square$

**Proposition 1.** *Assume the probability density functions of the noise terms $\{e(s, a)\}$ are continuous. The stochastic Bellman operators defined by Eq. (3) and Eq. (4) have approximated fixed points defined as Definition 1.*

*Proof.* Let $f(V) = \mathbb{E}[\widetilde{\mathcal{T}}V]$ denote the expected return of a stochastic Bellman operation. This function is continuous because all involved formulas only contain elementary functions. The given statement is proved by combining Lemma 1, 2, and 3. $\quad\square$

### A.3 THE INDUCED POLICY OF DOUBLE Q-LEARNING

**Proposition 3.** *Assume the noise terms $e_1$ and $e_2$ are independently generated in the double estimator stated in Eq. (4). Every approximated fixed point $V$ is equal to the ground truth value function $V^{\tilde{\pi}}$ with respect to an induced policy $\tilde{\pi}$.*

*Proof.* Let $V$ denote an approximated fixed point under the stochastic Bellman operator $\widetilde{\mathcal{T}}$ defined by Eq. (4). By plugging the definition of the induced policy into the stochastic operator of double Q-learning, we can get

$$\begin{aligned}
V(s) &= \mathbb{E}[\widetilde{\mathcal{T}}V(s)] \\
&= \mathbb{E}\left[(\widetilde{\mathcal{T}}Q_2)\left(s, \; \arg\max_{a \in \mathcal{A}}(\widetilde{\mathcal{T}}Q_1)(s, a)\right)\right] \\
&= \mathbb{E}\left[(\widetilde{\mathcal{T}}Q_2)\left(s, \; \arg\max_{a \in \mathcal{A}}((\mathcal{T}Q_1)(s, a) + e_1(s, a))\right)\right] \\
&= \mathbb{E}_{a \sim \tilde{\pi}(\cdot|s)}\left[(\widetilde{\mathcal{T}}Q_2)(s, a)\right] \\
&= \mathbb{E}_{a \sim \tilde{\pi}(\cdot|s)}\left[R(s, a) + \gamma \mathbb{E}_{s' \sim P(\cdot|s,a)} V(s') + e_2(s, a)\right] \\
&= \mathbb{E}_{a \sim \tilde{\pi}(\cdot|s)}\left[R(s, a) + \gamma \mathbb{E}_{s' \sim P(\cdot|s,a)} V(s')\right],
\end{aligned}$$

which matches the Bellman expectation equation. $\quad\square$

As shown by this proposition, the estimated value of a non-optimal fixed point is corresponding to the value of a stochastic policy, which revisits the incentive of double Q-learning to underestimate true maximum values.

## A.4 A SUFFICIENT CONDITION FOR MULTIPLE FIXED POINTS

**Proposition 2.** *Let $f_s\left(\{(\mathcal{T}Q)(s,a)\}_{a\in\mathcal{A}}\right) = \mathbb{E}[(\widetilde{\mathcal{T}}V)(s)]$ denote the expected output value of the stochastic Bellman operator $\widetilde{\mathcal{T}}$ on state $s$, and assume $f_s(\cdot)$ is differentiable. If a stochastic Bellman operator $\widetilde{\mathcal{T}}$ satisfies Eq. (5), there exists an MDP such that $\widetilde{\mathcal{T}}$ has multiple fixed points.*

$$\exists i, \ \exists X \in \mathbb{R}^{|\mathcal{A}|}, \quad \frac{\partial}{\partial x_i} f_s(X) > 1, \tag{5}$$

*where $X = \{x_i\}_{i=1}^{|\mathcal{A}|}$ denotes the input of the function $f_s$.*

*Proof.* Suppose $f_s$ is a function satisfying the given condition, and $x_i = \bar{x}$ and $X$ denote the corresponding point satisfying Eq. (5).

Let $g(x)$ denote the value of $f_s$ while only changes the input value of $x_i$ to $x$, so that $g'(\bar{x}) > 1$.

Since $f_s$ is differentiable, we can find a small region $\bar{x}_L < \bar{x} < \bar{x}_R$ around $\bar{x}$ such that $\forall x \in [\bar{x}_L, \bar{x}_R]$, $g'(x) > 1$. And then, we have $g(\bar{x}_R) - g(\bar{x}_L) > \bar{x}_R - \bar{x}_L$.

Consider to construct an MDP with only one state. We can use the action corresponding to $x_i$ to construct a self-loop transition with reward $r$. All other actions lead to a termination signal and an immediate reward where the immediate rewards correspond to other components of $X$. By setting the discount factor as $\gamma = \frac{\bar{x}_R - \bar{x}_L}{g(\bar{x}_R) - g(\bar{x}_L)} < 1$ and the reward as $r = \bar{x}_L - \gamma g(\bar{x}_L) = \bar{x}_R - \gamma g(\bar{x}_R)$, we can find both $\bar{x}_L$ and $\bar{x}_R$ are solutions of the equation $x = r + \gamma g(x)$, in which $g(\bar{x}_L)$ and $g(\bar{x}_R)$ correspond to two fixed points of the constructed MDP. $\qquad\square$

## A.5 A BAD CASE FOR CLIPPED DOUBLE Q-LEARNING

The stochastic Bellman operator corresponding to clipped double Q-learning is stated as follows.

$$\forall i \in \{1,2\}, \quad Q_i^{(t+1)}(s,a) = R(s,a) + \gamma \mathop{\mathbb{E}}_{s'\sim P(\cdot|s,a)}\left[V^{(t)}(s')\right] + e_i^{(t)}(s,a),$$
$$V^{(t)}(s) = \min_{i\in\{1,2\}} Q_i^{(t)}\left(s, \arg\max_{a\in\mathcal{A}} Q_1^{(t)}(s,a)\right). \tag{7}$$

An MDP where clipped double Q-learning has multiple fixed points is illustrated as Figure 5.

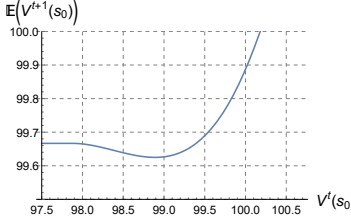

| $V(s_0)$ | $V(s_1)$ |
|----------|----------|
| 100.491 | 100.0 |
| 101.833 | 100.0 |
| 101.919 | 100.0 |

$R_{0,0}=1.35$

$R_{0,1}=100$

$s_0 \longrightarrow$ done

$\epsilon = 1.0 \quad \gamma = 0.99$

(a) A simple construction  (b) Numerical solutions of fixed points  (c) Visualizing non-monotonicity

Figure 5: (a) A simple MDP where clipped double Q-learning stated as Eq. (7) has multiple approximated fixed points. $R_{i,j}$ is a shorthand of $R(s_i, a_j)$. (b) The numerical solutions of the fixed points produced by clipped double Q-learning in the MDP presented above. (c) The relation between the input state-value $V^{(t)}(s_0)$ and the expected output state-value $\mathbb{E}[V^{(t+1)}(s_0)]$ generated by clipped double Q-learning in the constructed MDP.

# B    EXPERIMENT SETTINGS

## B.1    EVALUATION SETTINGS

All curves presented in this paper are plotted from the median performance of 5 runs with random initialization. The shaded region indicates 60% population around median. The evaluation is processed in every 50000 timesteps. Every evaluation point is averaged from 5 trajectories. The evaluated policy is combined with a 0.1% random execution.

## B.2    HYPER-PARAMETERS

All algorithm investigated in this paper use the same set of hyper-parameters.

- Number of *noop* actions while starting a new episode: 30;
- Number of stacked frames in observations: 4;
- Scale of rewards: clipping to $[-1, 1]$;
- Buffer size: $10^6$;
- Batch size: 32;
- Start training: after collecting 20000 transitions;
- Training frequency: 4 timesteps;
- Target updating frequency: 8000 timesteps;
- $\epsilon$ decaying: from $1.0$ to $0.01$ in the first $250000$ timesteps;
- Optimizer: Adam with $\varepsilon = 1.5 \cdot 10^{-4}$;
- Learning rate: $0.625 \cdot 10^{-4}$.

# C   ADDITIONAL EXPERIMENTS ON STANDARD ATARI BENCHMARK TASKS

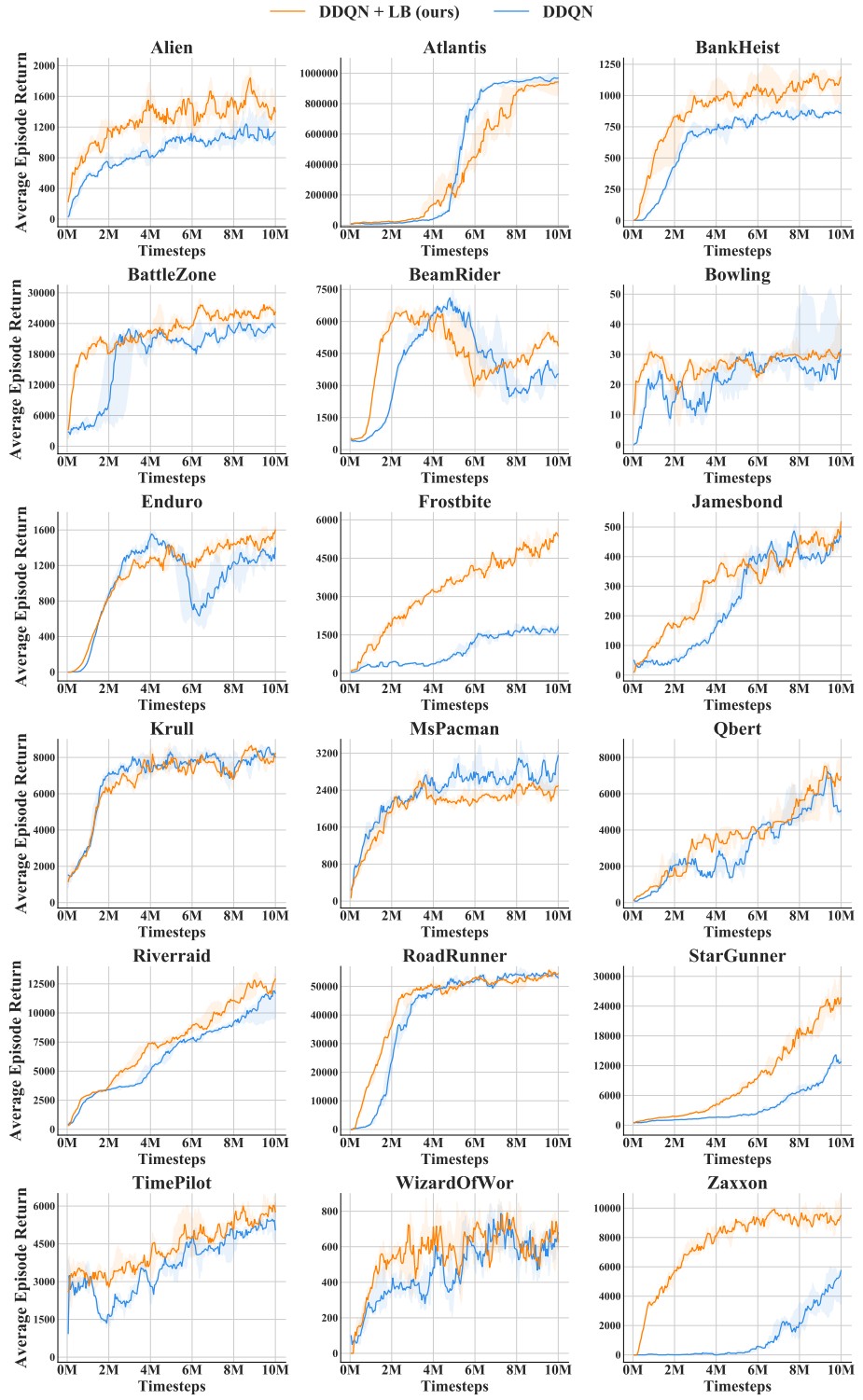

Figure 6: Learning curves on a suite of Atari benchmark tasks for comparing DDQN with or without lower bounded objective.

# D ABLATION STUDIES ON MULTI-STEP BOOTSTRAPPING

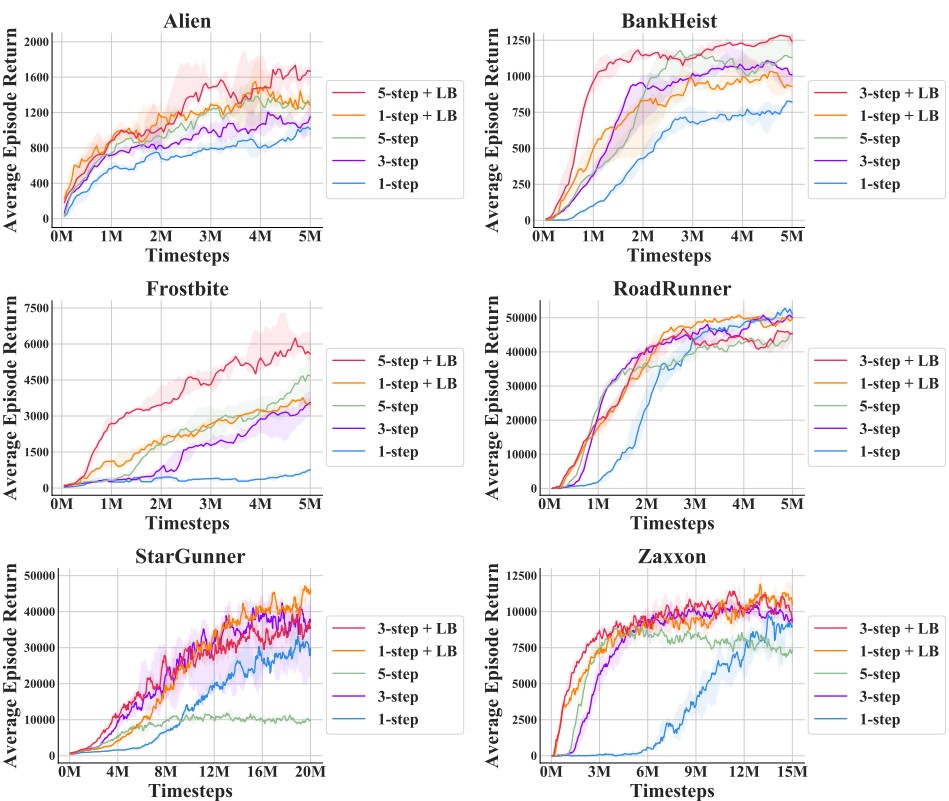

Figure 7: Learning curves on a suite of Atari benchmark tasks for comparing DDQN with $n$-step bootstrapping and the proposed lower bounded objective.

