# OpenReview forum: "On the Estimation Bias in Double Q-Learning"
_ICLR.cc/2021/Conference — Reject_

### Official Review · AnonReviewer2 · 2020-10-27

**Rating:** 6
**Confidence:** 4

**Review:**

This submission focused on the double Q-learning and investigated its underestimation bias issue. The authors claimed that using a double estimator can lead to multiple fixed points. To alleviate this issue, the authors proposed a correction approach by using the lower bound w.r.t. the real return. Finally, experiments on Atari games were conducted to show the improved performance, when compared with double DQN.

The authors aimed to analyze the underestimation issue in double Q-learning. The observation of multiple fixed points under the stochastic Bellman operator is interesting. The authors also tried to give more interpretation by providing sufficient conditions when this phenomenon happens. The proposed lower bounded double Q learning looks novel and reasonable.

I am a bit confused about the authors' explanation on the multiple fixed points in Section 3.2. The authors claimed in the last paragraph that "..due to the existence of the approximation error, the induced policy π˜ suffers from a remarkable uncertainty in determining the best action on state s0, which is the main cause of these extra fixed points. " If this is indeed the case, the multiple fixed points are not necessarily linked to double Q-learning, as the approximation error can appear in any form of function approximation.

Following the above point, I also don't quite understand the motivation about Definition 3. The authors claimed in Page 5 that "The monotonicity property is not a sufficient condition for Eq. (5) but it usually becomes a source of risks in practice." I just do not see how Definition 3 is necessarily connected with Proposition 3. The authors provided empirical observation that "During this process, a large magnitude of the first-order derivative could be found to meet the condition stated in Eq. (5)" However, this part in the curve also corresponds to the case of underestimation vanishing. Is this a contradiction here?

Another concern I have is regarding the comparison w.r.t. multi-step learning: The authors plot in Figure 3 the performance of n-step bootstrapping vs. LB, which I really appreciate. Note that this comparison is for the game Alien only; therefore, I am wondering if the conclusion holds for other games? It would be more convincing if the authors can provide more experimental results, as it can reinforce the authors' claim that the lower bounded approach rather than multi step learning is the main factor for performance improvement.

There is a related work on correcting biases in DDQN as follows, which may need to be discussed as well. The authors there also showed the improved performance on Atari games.

Song Z, Parr R, Carin L. Revisiting the softmax bellman operator: New benefits and new perspective, ICML 2019.

Minor comments: "...are meet..." -> "...are met..." in Page 5.

---

> ### Author Response · Authors · 2020-11-23
> **Response to Reviewer 2**
>
> Thanks for the constructive comments and an additional related work. In the revision, we included a short discussion of softmax bellman operator in the related work section. Also, we reorganized the discussion in section 3.2 to address the reviewer's concerns. We provide detailed clarification to your questions and concerns as below.
>
> **Q1**: Explanation on the multiple fixed points in Section 3.2. The authors claimed in the last paragraph that "... " If this is indeed the case, the multiple fixed points are not necessarily linked to double Q-learning, as the approximation error can appear in any form of function approximation.
>
> We have refined Section 3.2 to make the explanation more clear and rigorous. Our motivation of including this discussion is to explain why a non-optimal value function can become a fixed point under the stochastic Bellman operator.
>
> Given the fact that double Q-learning provide an unbiased estimation to the one-step target value of the learned policy (Hasselt, 2010), it is natural to consider why an unbiased policy evaluation would lead to non-optimal solutions. Regrading this question, we argue that, although the policy evaluation is unbiased, its variance would have negative effects on policy improvement. Intuitively speaking, due to the effects of approximation error, the policy would suffer from a remarkable uncertainty in determining the best action on the given state.
>
> In addition, this paper aims to establish a characterization of non-optimal fixed points in double Q-learning algorithms, but we do not claim that the issue of multiple fixed points only happens to double Q-learning and its variants. Although it is unknown whether there are also multiple fixed points in vanilla Q-learning, our proposed lower bound method suggests that the issue of non-optimal fixed points is easier to be fixed in double Q-learning, since it is more tractable to figure out underestimated values than overestimated ones.
>
> **Q2**: The motivation about Definition 3 of monotonicity. The authors provided empirical observation that "During this process, a large magnitude of the first-order derivative could be found to meet the condition stated in Eq. (5)" However, this part in the curve also corresponds to the case of underestimation vanishing. Is this a contradiction here?
>
> We introduce the concept of monotonicity and show the non-monotonicity of double Q-learning. This non-monotonicity is important for understanding the existence of multiple non-optimal fixed points in the constructed example. As discussed in the paper, although the non-monotonicity is not the sufficient condition to imply the existence of multiple fixed points, it gives a concrete and intuitive explanation for the curve presented in Figure 1c.
>
> Regarding the questions in Q1 and Q2, we reorganized the discussion in section 3.2 to make the presentation more clear. The formal theoretical statements are not changed.
>
> **Q3**: The comparison w.r.t. multi-step learning
>
> We included additional experiments on comparing our proposed lower bounded objective with multi-step learning, in Appendix D of the revision. The results support the claim stated in the paper. The bootstrapping horizon $n$ in multi-step learning is a very sensitive parameter, and an inappropriate choice of $n$ may even hurt the performance. In comparison, the proposed lower-bounded objective in our algorithm is more robust, and its improvement is compatible with the advantage of $n$-step bootstrapping.
>
> [1] *Hasselt HV. Double Q-learning. In Advances in neural information processing systems 2010 (pp. 2613-2621).*

---

### Official Review · AnonReviewer4 · 2020-10-28
**This paper analysed the underestimation bias induced by approximation error, by formalizing the underlying approximation, they theoretically proved the existence of multiple approximated fixed points which causes the converging to non-optimal solution. Besides, they proposed the lower bound double q-learning to overcome the underestimation bias.**

**Rating:** 6
**Confidence:** 3

**Review:**

This paper analysed the underestimation bias induced by approximation error, by formalizing the underlying approximation, they theoretically proved the existence of multiple approximated fixed points which causes the converging to non-optimal solution. Besides, they proposed the lower bound double q-learning to overcome the underestimation bias.

Strengths:
1.The paper is well-organized and the algorithm is simple yet effective.

2.The thought of formalizing the approximation error into noise is delighted which provides a way to theoretically analyze the effect of such error.

3.The definition and existence prove of approximated fixed points would helps the algorithm design to overcome the underestimation bias.

Weakness:
1.Some experiment is hard to understand. Table1 shows the TD-error and the absolute state-action value which didn’t demonstrate the small approximation error would cause significant estimation error which would cause the sub-optimal fixed points.

2.The effectiveness of lower bound double q-learning is doubtful. In MsPacman of Figure2, the algorithm shows slight performance decrease of Clipped DDQN, in some environment such as WizardOfWor, Zaxxon RoadRunner and BattleZone, these algorithms seems converge into same solutions. Besides, the algorithm would cause the overestimate the true maximum value.

---

> ### Author Response · Authors · 2020-11-23
> **Response to Reviewer 4 (Part 1)**
>
> Thanks for the comments. We provide clarification to your questions and concerns as below.
>
> **Q1**: Table1 shows the TD-error and the absolute state-action value which didn’t demonstrate the small approximation error would cause significant estimation error which would cause the sub-optimal fixed points.
>
> Table 1 is presented to demonstrate the existence of a constant approximation error which cannot be eliminated through training. It is intractable to directly visualize its effects on the sub-optiomal fixed points in an Atari game. Therefore, we illustrate the magnitude of absolute TD-error and argue that it is sufficient to raise the concerns on the potential issue identified by our analysis.
>
> **Q2**: In MsPacman of Figure2, the algorithm shows slight performance decrease of Clipped DDQN, in some environment such as WizardOfWor, Zaxxon RoadRunner and BattleZone, these algorithms seems converge into same solutions.
>
> Reducing biases is expected to improve asymptotic performance in most cases but not all cases. In cases where the optimal action has a significant advantage over sub-optimal ones, a small bias may not affect the action selection. In addition, the effects of estimation bias is known to be environment-dependent (Lan et al., 2020) since the bias may change the exploration behaviors. That is why the performance has a slight decrease in MsPacman.
>
> Besides the asymptotic performance, improving sample efficiency is another benefit of reducing the underestimation bias of double Q-learning. Empirical results show that our method learns more efficiently than baselines in most cases, as shown in Figure 2. The fixed points characterized by our analysis is similar to the concept of "saddle point" in the literature of optimization. A gradient step generated by a lucky mini-batch can help the algorithm to escape such saddle points. The proposed lower-bounded objective is a heuristics to provide the incentive for quickly escaping non-optimal saddle points. We include this discussion in section 3.2 of the revision.
>
> [1] *Lan Q, Pan Y, Fyshe A, White M. Maxmin Q-learning: Controlling the Estimation Bias of Q-learning. International Conference on Learning Representations 2020.*

---

> > ### Author Response · Authors · 2020-11-23
> > **Response to Reviewer 4 (Part 2)**
> >
> > **Q3**: The algorithm would cause the overestimate the true maximum value.
> >
> > It is possible for our method to suffer from overestimation, especially in stochastic environments. Actually, our algorithm exploits this possibility and considers a trade-off between such overestimation of the trajectory-based lower bound and the underestimation of double Q-learning. To our best knowledge, there is no existing approach can obtain fully unbiased value estimation in Q-learning methods. Most of related works also focus on the trade-off between different sources of biases (Fujimoto et al., 2018; Lan et al., 2020).
> >
> > The effectiveness of the trade-off induced by our algorithm can be actually highlighted in stochastic environments. In the other hand, our method excels at addressing the issue of non-optimal fixed points, which is exacerbated by the environment stochasticity. This is because the environment stochasticity increases the difficulty of optimization in the target regression and leads to larger approximation error that generates non-optimal fixed points.
> >
> > In the revision, we evaluate our proposed method in a variant of Atari benchmark task with stronger stochasticity (Machado et al, 2018), in which every environment step may ignore the given instruction and repeat the previous action with probability 0.25. In this experiment, we include two additional baselines, Rainbow (Hessel et al., 2018) and C51 (Bellemare et al., 2017), provided by Dopamine open-source implementation (Castro et al., 2018). To fairly compare with these state-of-the-art baselines that integrates many standard techniques, our proposed lower bounded objective is combined with a standard technique of multi-step bootstrapping. Our method provides a performance improvement and significantly outperforms baselines, as shown in Figure 4 in the revision.
> >
> > [2] *Fujimoto S, Hoof H, Meger D. Addressing Function Approximation Error in Actor-Critic Methods. In International Conference on Machine Learning 2018 Jul 3 (pp. 1587-1596).*
> >
> > [3] *Machado MC, Bellemare MG, Talvitie E, Veness J, Hausknecht M, Bowling M. Revisiting the arcade learning environment: Evaluation protocols and open problems for general agents. Journal of Artificial Intelligence Research. 2018 Mar 19;61:523-62.*
> >
> > [4] *Hessel M, Modayil J, van Hasselt H, Schaul T, Ostrovski G, Dabney W, Horgan D, Piot B, Azar MG, Silver D. Rainbow: Combining Improvements in Deep Reinforcement Learning. In AAAI 2018 Jan 1.*
> >
> > [5] *Bellemare MG, Dabney W, Munos R. A Distributional Perspective on Reinforcement Learning. In International Conference on Machine Learning 2017 Jul 17 (pp. 449-458).*
> >
> > [6] *Castro PS, Moitra S, Gelada C, Kumar S, Bellemare MG. Dopamine: A research framework for deep reinforcement learning. arXiv preprint arXiv:1812.06110. 2018 Dec 14.*

---

### Official Review · AnonReviewer3 · 2020-10-28
**An interesting topic that needs to be better understood**

**Rating:** 3
**Confidence:** 4

**Review:**

The paper considers the problem of learning a value function in a deterministic MDP and proposes a heuristic to reduce the bias of value estimates under the greedy policy. For this, they consider the class of Double Q-learning algorithms and describe a setting where estimation under this algorithm can lead to multiple fixed point solutions. They propose to control underestimation bias by estimating the next-state value as the maximum of either the current next-state value or the value of some trajectory in the replay buffer. They also propose another version that uses clipped target values. The two algorithms are evaluated on several Atari games and show, in some cases, improved transient behavior over double DQN.

Controlling for bias due to function approximation is an important problem, and one where general solutions could have a potentially large impact for RL and AI. Giving this subject the attention it deserves will require an understanding of the bias problem in stochastic settings. Many will be curious about what can be said about this problem and its solution(s) using formal mathematical language. The multiple-solutions hypothesis is a plausible approach to gaining a better understanding of bias, but there could also be other angles to explore.

I think the current research this paper presents should be broadened to understand not just deterministic environments, but stochastic ones as well. The experimental results should also be questioned: why is there little to no asymptotic performance gain using the proposed method if it indeed produces an estimate with less bias? There are more questions the community would wish to answer with a paper like this; Where does underestimation bias come from, and how can it be controlled in general? Currently, however, the presentation and technical material are not sharp enough to answer such questions. Therefore, it is important that the authors continue to mature this work so that, when it is ready for publication, it will make a significant contribution.

---

> ### Author Response · Authors · 2020-11-23
> **Response to Reviewer 3 (Part 1)**
>
> Thanks for the comments. We provide clarification to your questions and concerns as below.
>
> **Q1**: The current research this paper presents should be broadened to understand not just deterministic environments, but stochastic ones as well.
>
> The formal analysis part of this paper does not make any assumption of deterministic dynamics. We present a deterministic MDP as an example only under the considerations of simplification. The revealed issues of double Q-learning definitely can generalize to stochastic environments.
>
> The design of the proposed lower-bounded objective is inspired by deterministic environments, but it can also be applied to stochastic environments. When the environment carries some extent of stochasticity, the trajectories with high return values would cause additional risks of overestimation through the proposed lower bounded objective. In the other hand, our method excels at addressing the issue of non-optimal fixed points, which is exacerbated by the environment stochasticity. This is because the environment stochasticity increases the difficulty of optimization in the target regression and leads to larger approximation error that generates non-optimal fixed points.  From this perspective, our method is a trade-off between the underestimation bias of double Q-learning and the overestimation of the trajectory-based lower-bound estimation.
>
> In the revision, we evaluate our proposed method in a variant of Atari benchmark task with stronger stochasticity (Machado et al, 2018), in which every environment step may ignore the given instruction and repeat the previous action with probability 0.25. In this experiment, we include two additional baselines, Rainbow (Hessel et al., 2018) and C51 (Bellemare et al., 2017), provided by Dopamine open-source implementation (Castro et al., 2018). As shown in Figure 4 in the revision, our method can provide a performance improvement and significantly outperform baselines, which demonstrates the effectiveness of the bias trade-off induced by our method.
>
> **Q2**: Why is there little to no asymptotic performance gain using the proposed method if it indeed produces an estimate with less bias?
>
> We would like to clarify three points regarding the comparison of asymptotic performance.
>
> 1. When applying the proposed lower-bounded objective to DDQN, there are roughly half of environments can get benefits in terms of the asymptotic performance. Please refer to Figure 5 in Appendix for a more clear comparison between DDQN w/ and wo/ the lower-bounded objective.
>
> 2. Reducing biases is expected to improve asymptotic performance in most cases but not all cases. When the optimal action has a significant advantage over sub-optimal ones, a small bias cannot affect the action selection.
>
> 3. Besides the asymptotic performance, improving sample efficiency is also a benefit of reducing the underestimation bias of double Q-learning. The fixed points characterized by our analysis is similar to the concept of "saddle point" in the literature of optimization. A gradient step generated by a lucky mini-batch can help the algorithm to escape such saddle points. The proposed lower-bounded objective is a heuristics to provide the incentive for escaping non-optimal saddle points. We include this discussion in section 3.2 of the revision.
>
> [1] *Machado MC, Bellemare MG, Talvitie E, Veness J, Hausknecht M, Bowling M. Revisiting the arcade learning environment: Evaluation protocols and open problems for general agents. Journal of Artificial Intelligence Research. 2018 Mar 19;61:523-62.*
>
> [2] *Hessel M, Modayil J, van Hasselt H, Schaul T, Ostrovski G, Dabney W, Horgan D, Piot B, Azar MG, Silver D. Rainbow: Combining Improvements in Deep Reinforcement Learning. In AAAI 2018 Jan 1.*
>
> [3] *Bellemare MG, Dabney W, Munos R. A Distributional Perspective on Reinforcement Learning. In International Conference on Machine Learning 2017 Jul 17 (pp. 449-458).*
>
> [4] *Castro PS, Moitra S, Gelada C, Kumar S, Bellemare MG. Dopamine: A research framework for deep reinforcement learning. arXiv preprint arXiv:1812.06110. 2018 Dec 14.*

---

> > ### Author Response · Authors · 2020-11-23
> > **Response to Reviewer 3 (Part 2)**
> >
> > **Q3**: Where does underestimation bias come from, and how can it be controlled in general?
> >
> > Double Q-learning introduces an underestimation bias, which is proved by the work (Hasselt, 2010). When the action selection is decided upon a noisy value estimation, the policy may become non-optimal due to the effects of estimation error. Although double Q-learning uses a cross validation to obtain an unbiased policy evaluation, the non-optimal action selection would lead to an underestimation of the true maximum value. The multiple fixed points phenomenon discussed by this paper is a novel characterization on the effects of such underestimation bias.
> >
> > The issue of estimation bias has been investigated for 27 years (Thrun and Schwartz, 1993). There is no existing approach can obtain fully unbiased value estimation in Q-learning methods. Most of related works focus on the trade-off between different sources of biases (Fujimoto et al., 2018; Lan et al., 2020). One novelty of this paper is a new trade-off between the overestimation of the trajectory-based lower bound and the underestimation of double Q-learning. Please also refer to Q1.
> >
> > [5] *Hasselt HV. Double Q-learning. In Advances in neural information processing systems 2010 (pp. 2613-2621).*
> >
> > [6] *Thrun S, Schwartz A. Issues in using function approximation for reinforcement learning. In Proceedings of the 1993 Connectionist Models Summer School Hillsdale, NJ. Lawrence Erlbaum 1993 Dec.*
> >
> > [7] *Fujimoto S, Hoof H, Meger D. Addressing Function Approximation Error in Actor-Critic Methods. In International Conference on Machine Learning 2018 Jul 3 (pp. 1587-1596).*
> >
> > [8] *Lan Q, Pan Y, Fyshe A, White M. Maxmin Q-learning: Controlling the Estimation Bias of Q-learning. International Conference on Learning Representations 2020.*

---

### Official Review · AnonReviewer1 · 2020-10-28
**simple yet effective approach to address estimation bias**

**Rating:** 6
**Confidence:** 3

**Review:**

##################################

Summary: This paper investigates the effects of approximation error in Q-learning - more specifically, the problem of having multiple non-optimal fixed points in double Q-learning. The authors claim that double Q-learning, which is a well-known approach to alleviate overestimation bias of Q-learning, can suffer from underestimation bias, and can lead to non-optimal fixed points. This paper provides theoretical evidence to support this claim. The main idea of this paper is to add real returns as a lower bound to the objective, and alleviate underestimation bias. Empirical results in Atari domains are presented.

################################################

Pros

1. This paper addresses an important problem in reinforcement learning - approximation errors and bias in Q-learning. The major focus of the RL community on estimation bias has been overestimation bias (van Hasselt 2010), but this paper brings up the issue of underestimation bias (Lan et al 2020).

2. This paper formalizes a way to model approximation error in Bellman operators in Q-learning (by setting a random noises as approximation error), and derives multiple theoretical insights and propositions (Existence of multiple fixed points in double q-learning).

3. The idea of having return signal V_\tau (the returns gained from real trajectories) as a lower bound is simple and easy to implement. As for V(frozen\theta), the authors use both DDQN and clipped-DDQN objective; they combined these two with the lower-bound objective, and showed the performance of DDQN-LB and clipped-DDQN-LB., which performs better than previous baselines in multiple Atari domains.

4. Comparison with n-step bootstrapping (Figure3) is also helpful; the idea is compatible with n-step bootstrapping, so it can be combined to improve the performance.

################################################



Questions

1. Section 4: Choice of trajectory \tau for return signal V_\tau(s_{t+1}): this term is the discounted return of trajectory \tau in the replay buffer. Is this just a return value of one trajectory? How do you address the stochasticity of having real return value of one trajectory? Or is this an average of multiple trajectories that start from s_{t+1}?

2. Have you ever considered other baselines like duel-DQN or Rainbow to compare the performance of your approach? I don’t think the empirical results (comparison with DDQN, DDQN, cDDQN, maxmin DQN, etc)  are already sufficient, but I’m just curious how it performs well compared to SOTA baselines.


###################################################

Reasons for score: I think this paper addresses an important problem in RL, and presents a simple yet effective approach to address this problem. This paper is also theoretically well supported.

---

> ### Author Response · Authors · 2020-11-23
> **Response to Reviewer 1**
>
> Thanks for the comments. We provide clarification to your questions and concerns as below.
>
> **Q1**: This term is the discounted return of trajectory $\tau$ in the replay buffer. Is this just a return value of one trajectory? How do you address the stochasticity of having real return value of one trajectory? Or is this an average of multiple trajectories that start from $s_{t+1}$?
>
> In our implementation, the return value is estimated by one trajectory. This estimation is reliable in deterministic environments. When the environment carries some extent of stochasticity, the trajectories with high return values would cause additional risks of overestimation through the lower bounded target. In the other hand, our method excels at addressing the issue of non-optimal fixed points, which is exacerbated by the environment stochasticity. This is because the environment stochasticity increases the difficulty of optimization in the target regression and leads to larger approximation error that generates non-optimal fixed points. From this perspective, our method is a trade-off between the underestimation bias of double Q-learning and the overestimation of the trajectory-based lower-bound estimation.
>
> In the revision, we evaluate our proposed method in a variant of Atari benchmark task with stronger stochasticity (Machado et al, 2018), in which every environment step may ignore the given instruction and repeat the previous action with probability 0.25. Empirical results show that our proposed method can outperform state-of-the-art baselines such as Rainbow (Hessel et al., 2018) and C51 (Bellemare et al., 2017), which demonstrates the effectiveness of the bias trade-off induced by our method. Please also refer to Q2 for details.
>
> **Q2**: Have you ever considered other baselines like duel-DQN or Rainbow to compare the performance of your approach?
>
> In the revision, we also conduct experiments in the sticky-action (i.e., stochastic) Atari environment (Machado et al, 2018) in addition to the standard one, and include more baselines for comparison.
>
> In the standard Atari benchmark tasks, we investigate two additional baselines, dueling DDQN (Wang et al., 2016) and Averaged DQN (Anschel et al., 2017). Averaged DQN is a related work which aims to address the approximation error by retaining multiple copies of target networks. Our proposed method outperforms them in all 9 environments, as shown in Figure 2 in the revision.
>
> In the Atari benchmark tasks with sticky actions, we include two additional baselines, Rainbow (Hessel et al., 2018) and C51 (Bellemare et al., 2017), provided by Dopamine open-source implementation (Castro et al., 2018). Both of them use a distributional value representation to address the optimization difficulty caused by environment stochasticity. To fairly compare with these state-of-the-art baselines that integrates many standard techniques, we also combine our proposed lower bounded objective with multi-step bootstrapping. Our method provides a performance improvement and significantly outperforms baselines, as shown in Figure 4 in the revision.
>
> [1] *Wang Z, Schaul T, Hessel M, Hasselt H, Lanctot M, Freitas N. Dueling network architectures for deep reinforcement learning. In International conference on machine learning 2016 Jun 11 (pp. 1995-2003).*
>
> [2] *Anschel O, Baram N, Shimkin N. Averaged-dqn: Variance reduction and stabilization for deep reinforcement learning. In International Conference on Machine Learning 2017 Jul 17 (pp. 176-185). PMLR.*
>
> [3] *Hessel M, Modayil J, van Hasselt H, Schaul T, Ostrovski G, Dabney W, Horgan D, Piot B, Azar MG, Silver D. Rainbow: Combining Improvements in Deep Reinforcement Learning. In AAAI 2018 Jan 1.*
>
> [4] *Bellemare MG, Dabney W, Munos R. A Distributional Perspective on Reinforcement Learning. InInternational Conference on Machine Learning 2017 Jul 17 (pp. 449-458).*
>
> [5] *Castro PS, Moitra S, Gelada C, Kumar S, Bellemare MG. Dopamine: A research framework for deep reinforcement learning. arXiv preprint arXiv:1812.06110. 2018 Dec 14.*

---

### Public Comment · ~Yasuhiro_Fujita1 · 2020-11-11
**On the lowerbounding trick**

For your information, if I understand the proposed lowerbounding trick correctly, we used the same trick in our paper [1], which has been accepted at IROS 2020 (see Paragraph e of Section IV.A). To be clear, we did it for efficient learning in sparse-reward multi-goal settings, not for correcting the bias in double Q-learning, which is a difference. We applied it to goal-conditioned QT-Opt, where we did not use the double Q-learning trick.

[1] Y. Fujita, K. Uenishi, A. Ummadisingu, P. Nagarajan, S. Masuda, and M. Y. Castro, “Distributed Reinforcement Learning of Targeted Grasping with Active Vision for Mobile Manipulators,” IROS, 2020. https://arxiv.org/abs/2007.08082

---

> ### Author Response · Authors · 2020-11-23
> **Thanks for your comment.**
>
> Dear Yasuhiro,
>
> Thanks for your comment. We have included discussions of your mentioned paper in related work of our revision.
>
> As you mentioned, we aim to address different problems by introducing the lower bound trick. In your work, the lower bound trick helps to address the slow value propagation caused by the sparse-reward setting. In our work, double Q-learning has a natural incentive to underestimate values, whose effects could be reduced by a lower bounded target.
>
> By the way, we found that, in the original paper of QT-Opt (Appendix A, https://arxiv.org/abs/1806.10293), they adopted clipped double Q-learning to improve the performance. We are curious why you did not use clipped double Q-learning.
>
> Thanks,
>
> Authors

---

> > ### Comment · ~Yasuhiro_Fujita1 · 2021-01-18
> > **> clipped double Q-learning**
> >
> > Thank you for including the discussion of our paper!
> >
> > > By the way, we found that, in the original paper of QT-Opt (Appendix A, https://arxiv.org/abs/1806.10293), they adopted clipped double Q-learning to improve the performance. We are curious why you did not use clipped double Q-learning.
> >
> > I'm sorry I didn't make it clear enough. We used QT-Opt's clipped double Q-learning in the experiments for our paper. We took minimum of predictions by two target Q-networks, one with periodically synchronized parameters and one with Polyak-averaged parameters as described in Appendix F.4 in the QT-Opt paper.

---

### Decision · Program_Chairs · 2021-01-07
**Final Decision**

**Decision:**

Reject

**Comment:**

This paper is rejected.

I and the reviewers appreciate the changes made by the authors. The paper presents:
* An analysis (based on techniques from previous work) of double Q-learning which shows that in an analytic model, double Q-learning can have multiple sub-optimal "approximated" fixed points.
* Propose a modification of the update that uses collected trajectories to lower bound the optimal value.
* Experiments on several Atari games.

While the theoretical results on double Q-learning are interesting, the authors provide little theoretical analysis of their proposed approach. Doing so will significantly strengthen the paper. Additionally, reviewers had concerns about the experiments. R2 questions the parameter setting in the multi-step experiments.